# Laboratory Comparison of Low-Cost Particulate Matter Sensors to Measure Transient Events of Pollution

**DOI:** 10.3390/s20082219

**Published:** 2020-04-15

**Authors:** Florentin Michel Jacques Bulot, Hugo Savill Russell, Mohsen Rezaei, Matthew Stanley Johnson, Steven James Johnston Ossont, Andrew Kevin Richard Morris, Philip James Basford, Natasha Hazel Celeste Easton, Gavin Lee Foster, Matthew Loxham, Simon James Cox

**Affiliations:** 1Faculty of Engineering and Physical Sciences, University of Southampton, Southampton SO17 1BJ, UK; sjj698@zepler.org (S.J.J.O.); p.j.basford@soton.ac.uk (P.J.B.); s.j.cox@soton.ac.uk (S.J.C.); 2Southampton Marine and Maritime Institute, University of Southampton, Southampton SO16 7QF, UK; nhcs1g13@soton.ac.uk (N.H.C.E.); gavin.foster@noc.soton.ac.uk (G.L.F.); m.loxham@soton.ac.uk (M.L.); 3Danish Big Data Centre for Environment and Health (BERTHA), Aarhus University, DK-4000 Roskilde, Denmark; hugo.russell@envs.au.dk; 4Airlabs Denmark, Lersø Park Allé 107, DK-2100 Copenhagen Ø, Denmark; msj@chem.ku.dk; 5Department of Environmental Science, Atmospheric Measurement, Aarhus University, Frederiksborgvej 399, DK-4000 Roskilde, Denmark; 6Department of Chemistry, University of Copenhagen, Universitetsparken 5, DK-2100 Copenhagen Ø, Denmark; mohsen@chem.ku.dk; 7National Oceanography Centre, Southampton SO14 3ZH, UK; andmor@noc.ac.uk; 8School of Ocean and Earth Science, National Oceanography Centre, University of Southampton, Southampton SO14 3ZH, UK; 9Faculty of Medicine, University of Southampton, Southampton SO17 1BJ, UK; 10National Institute for Health Research, Southampton Biomedical Research Centre, Southampton SO16 6YD, UK; 11Institute for Life Sciences, University of Southampton, Southampton SO17 1BJ, UK

**Keywords:** air pollution, low-cost sensor, laboratory study, particulate matter

## Abstract

Airborne particulate matter (PM) exposure has been identified as a key environmental risk factor, associated especially with diseases of the respiratory and cardiovascular system and with almost 9 million premature deaths per year. Low-cost optical sensors for PM measurement are desirable for monitoring exposure closer to the personal level and particularly suited for developing spatiotemporally dense city sensor networks. However, questions remain over the accuracy and reliability of the data they produce, particularly regarding the influence of environmental parameters such as humidity and temperature, and with varying PM sources and concentration profiles. In this study, eight units each of five different models of commercially available low-cost optical PM sensors (40 individual sensors in total) were tested under controlled laboratory conditions, against higher-grade instruments for: lower limit of detection, response time, responses to sharp pollution spikes lasting <1 min, and the impact of differing humidity and PM source. All sensors detected the spikes generated with a varied range of performances depending on the model and presenting different sensitivity mainly to sources of pollution and to size distributions with a lesser impact of humidity. The sensitivity to particle size distribution indicates that the sensors may provide additional information to PM mass concentrations. It is concluded that improved performance in field monitoring campaigns, including tracking sources of pollution, could be achieved by using a combination of some of the different models to take advantage of the additional information made available by their differential response.

## 1. Introduction

Air pollution has been identified as the greatest environmental cause of morbidity and mortality in the world at present, and particulate matter (PM) is shown to have the greatest health impact of all measured pollutants [1,2]. This disease burden is chiefly attributed to diseases such as: asthma, chronic obstructive pulmonary disease (COPD), lung cancer, ischaemic heart disease, and stroke, but air pollution has more recently been associated with a host of other diseases and impacts, such as loss of cognitive performance, diabetes, Alzheimer’s and Parkinson’s disease [3,4,5,6,7]. In a recent study, Wu et al. [8] showed that small increase in PM_2.5_ long-term exposure is linked with large increased mortality from COVID-19.

The emission of PM also has wide-reaching climatic effects across the globe through particles acting as cloud-condensation nuclei, altering weather patterns and potentially contributing to droughts, or by lowering the Earth’s albedo, particularly as black carbon in snow, and inducing climate change [9,10].

Due to its range of impacts, accurate measurement of PM concentration in the environment is of great interest, as it is essential to enforcing adherence to legal regulations, determining the exact extent of PM pollution, understanding the exposure of individuals, and monitoring the efficacy of remediation efforts.

Current monitoring networks for regulatory purposes rely on a small number of sparsely placed, expensive monitoring stations. For instance, only 24 countries have more than three monitoring station per million inhabitants [11]. These stations provide accurate measurements, generally on an hourly basis, for their exact location but as hourly mean levels of pollutants vary considerably over tens of metres, this level of coverage cannot provide accurate exposure data for citizens [12,13,14,15]. The rapid variation in urban pollution levels is due to the strong influence of local sources and meteorological conditions, as well as street canyon effects in these areas [16]. The traditional monitoring strategy does not account for indoor exposure which can be greatly elevated relative to outdoor, for instance through use of solid-fuels for heating and cooking, emissions from the cooking itself, and household aerosols. Around half of all pollution related deaths are attributed to indoor pollution exposure although this cannot be accurately studied by outdoor monitoring stations [1,17]. Air pollution has a disproportionately large impact on low and middle income countries [1], where reference-grade air quality monitoring is typically absent, further highlighting the need for alternative ways to accurately quantify air pollution.

Low-cost PM sensors, generally based on light-scattering measurement techniques, have several benefits. They are easy to use compared to monitoring stations as they are significantly smaller and use much less power, and they are amenable to temporary and mobile installation. Their low cost has the overriding benefit that they can be deployed in great numbers which, added to their high frequency of measurement (≈1 Hz), increases the spatio-temporal granularity of available measurements. As such they can augment and improve the current system for monitoring urban air pollution [18,19]. Once they are validated there are myriad other uses for low-cost and portable PM sensors. In particular, a better understanding of the spatio-temporal variation of PM concentration would improve the accuracy of exposure classification in epidemiological studies, while a better understanding of personal exposure at the individual level may be required to be able to accurately identify biomarkers of PM exposure and effect [20]. Other examples of their diverse potentials include source tracking and early warning systems for forest fires that would combine CO, CO_2_ and PM monitors and drone-mounted sensors for monitoring shipping emissions [21,22]. They also have the potential to help in directing remediation efforts and to enable citizens to make informed decisions about how to manage their personal exposure, which could significantly lower their risk of disease even if pollution levels do not fall [23]. Low-cost PM sensors could also help improve the understanding of indoor air pollution, in particular where high levels of outdoor air pollution preclude ventilation with outdoor air [24,25]. At present there is insufficient data to validate their use for regulatory purposes [26,27,28], which is particularly urgent in developing countries.

Despite the multiple benefits of their low cost, low maintenance requirements, small size, small power requirements and portability, current low-cost sensors have important shortcomings which limit their potential, and must be addressed before their wide-scale accreditation and use. These include [29]: (1) a lower limit of detection which may be of the same order of magnitude than measured ambient PM concentrations, especially in developed nations; (2) variable responses to changes in humidity and temperature; (3) inability to measure PM mass directly; (4) lack of sensitivity to particles <0.3
μm in diameter; (5) differential response to particles depending on their composition; and (6) inter-unit variability even for units of the same model/manufacturer. While these issues are known there is not currently enough evidence to correct them, either from field or laboratory based studies. More information on the accuracy, precision, and reproducibility of the data, and how these confounding factors may differentially affect these, is necessary for improving the way in which the output of such sensors is used and interpreted. Morawska et al. [28] reviewed the use of low-cost sensors and concluded that low-cost sensors have “already changed the paradigm of air pollution monitoring” and are already suitable for supplementing ambient monitoring networks by providing data of lesser quality than the reference monitoring networks but higher spatio-temporal resolution [30]. The review also concludes that the sensors are suitable for engaging citizens, but are not yet of sufficient quality for source compliance or personal exposure monitoring.

The issues above mentioned are attributed to the sensing mechanism, which relies on Mie scattering [31], and due to this, the method works best for particle diameters larger than the wavelength of the incident light, typically a few hundred nanometers. This excludes ultrafine particles from the measurement, which may be key in driving adverse health effects, while only representing a small proportion of the mass of PM. Particles with a diameter greater than ca. 10 μm, because of their weight and size are harder to draw into the sensing area. Humidity and temperature fluctuations are known to alter the recorded particle mass as they can change the amount of water adsorbed by particles and therefore their recorded size but also modifies the refractive index of the particles. Low-cost PM sensors tend to over report above a given Relative Humidity (RH) (generally, around 80%) and if the particular conditions for condensation of water are reached this can further impact measurement [32,33]. Jayaratne et al. [32] notes that this behaviour may also affect reference instruments such as Tapered Element Oscillating Microbalances (TEOMs) to a certain extent. In Nyarku et al. [34] a mobile phone equipped with PM and Volatile Organic Compounds (VOC) sensors was tested against reference instruments with a range of pollutants, the PM sensor was found to have a linear response at elevated PM concentrations but not at the lower concentrations pertinent to ambient monitoring. The lower accuracy of the sensors at lower concentration has been observed by other studies [35,36]. Particle density and refractive index are assumed in the conversion of sensor signal to a mass of PM so the calibration of a sensor will only be accurate for PM of a specific density, as determined by composition and therefore the sensors must be re-calibrated for different environments [27]. In Sousan et al. [37] it was shown that this can be accounted for with a source-specific calibration when sensors are used for occupational exposure.

A number of recent studies of low-cost sensors for both static and mobile applications have employed manufacturer-supplied performance data without performing an independent calibration [28,38]. However, there is evidence that the deployment conditions will impact sensor response and that sensors exhibit inter-model variability [33,35]. Sayahi et al. [38] developed a calibration chamber for low-cost PM sensors that could identify malfunctioning sensors and expose inter-model inconsistencies as well as differing response to different aerosol types between sensors. This inter-model variability was also shown for real indoor pollution events across eight sensors in Zou et al. [39]. Similar results were found for low-cost sensors in the outdoor environment in Badura et al. [40], where a strong linear relationship to TEOM measurements was recorded for most of the sensors tested. Individual field calibration of the sensors may be required and a greater understanding of the effect of prevailing conditions on sensor response must be developed to address this issue. Laboratory and field conditions testing are complementary and are both required to fully assess the performances of the sensors [28]. Outdoor studies and calibration under realistic conditions need to be of sufficient duration to capture a range of environmental conditions and aerosol compositions. These studies are time consuming, can only capture local environmental conditions, and are less explanatory than controlled experiments. Laboratory characterisation enables fast evaluation of specific behaviour of the sensors [38,41,42].

The great majority of the recent studies investigating the performance of low-cost PM sensors in a laboratory environment, average concentrations over extended periods of time, for example 1 h, are measured, or pollution events which concentration then decreases slowly over several minutes or hours [41]. Other laboratory studies often tested a small number of sensors or sensors from a same manufacturer [32,34,37,43,44,45,46], and often employed high concentrations of PM, above 100 μg/m${}^{3}$ that are left to decrease through gravity or particle loss [34,36,44,46,47,48].

In this study, a large number of sensors from different manufacturers were evaluated concurrently. The sensors’ response, at a 10 s sampling period, to spikes of pollution lasting ≈1 min was monitored, with a PM_2.5_ median peak concentrations <40 μg/m${}^{3}$, from two different sources of pollution. Such peaks are similar to measurements undertaken under outdoor conditions by these sensors [35,40] and may inform regarding the potential and limitations of these sensors for outdoor deployments. This is especially pertinent for their use as personal exposure monitors or for mobile applications and especially to assess their capacity to track sources of pollution. This study aims: (1) to cross compare the performance of a range of commercially available low-cost PM sensors; (2) to better understand the response of these sensors to short events of pollution; (3) to characterise the influence of environmental factors; (4) to characterise inter-unit variability; (5) to demonstrate that low-cost PM sensors can be used, despite their limitations and with proper considerations, to track sources of pollution.

## 2. Materials and Methods

### 2.1. Low-Cost PM Sensors

In this study, five different models of low-cost PM sensors from different manufacturers have been tested [49]: Honeywell HPMA115S0, Alphasense OPC-R1, Novafitness SDS018, Sensirion SPS030 and Plantower PMS5003 (see Figure 1a). These models have been selected due to their frequency of use in air quality monitoring studies and for their compact size and price which makes them suitable for large scale deployments. The Plantower PMS5003 has been widely studied [32,35,40,43,45,50,51]. The Alphasense OPC-R1 is a lower cost and lower size version of the OPC-N2 which is widely used and studied [33,35,40,52,53]. To the best of the authors knowledge, the Honeywell HPMA115S0 has only been characterised by a limited number of studies [35] and the Novafitness SDS018 and the Sensirion SPS030 have not been characterised in the literature. The different models of sensor are shown in Figure 1b and their specifications are provided in Table 1. They were selected from a range of commercially available sensors from different manufacturers to get an idea of performance variation across range of models and compare some of the models widely tested in the literature to models new or less often tested. The sensors studied are all Optical Particle Counter (OPC) based on Mie theory of light-scattering. A built-in fan draws the particles in to the sensors. A beam of light then illuminates each particle as it passes through the light chamber and the scattered light beam is then recorded on a photodetector. The sensors convert the scattered light into an electrical signal to count and calculate the size of the particle which is then inferred as a mass concentration in real time. The sensors assume that the particles are spherical and all have fixed density and refractive index [31]. The Honeywell HPMA115S0 outputs PM_2.5_ and PM_10_ concentrations in $\mu g/m^{3}$ [54]. The Alphasense OPC-R1 measures concentration of PM_1_, PM_2.5_ and PM_10_. The particle size is recorded by the intensity of scattered light and classified at rates up to ≈10,000 particles/s. It provides particle size distribution, classifying the size distribution into 16 bins from 0.4–12.4 μm. Particle mass concentration is calculated from the particle size spectra assuming a density of 1.65 g/mL and a refractive index of 1.5. The Alphasense OPC-R1 is calibrated for particle count against another Alphasense OPC-R1 previously calibrated against a OPS TSI 3330 with monodisperse polystyrene latex microspheres of specific sizes [55]. The Novafitness SDS018 outputs PM_2.5_ and PM_10_ mass concentration [56]. The Sensirion SPS030 outputs PM_1_, PM_2.5_, PM${}_{4}$ and PM_10_ mass concentrations along with the associated particle size distribution. It has been calibrated for PM_2.5_ against a TSI DustTrak DRX 8533 using a defined particle distribution of potassium chloride [57]. For long-term monitoring, the Sensirion SPS030 includes a cleaning function, triggering the fan to work at maximal speed for 10 s.

The Plantower PMS5003 can be used to obtain the number of particles divided into six size bins, >0.3
μm, >0.5
μm, >1.0
μm, >2.5
μm, >5.0
μm and >10 μm. It then computes mass concentrations and outputs PM_1_, PM_2.5_ and PM_10_ mass concentrations. Equivalent particle diameter and the number of particles can be calculated by microprocessor based on Mie theory [50].

The sensors are placed within four different air quality monitors as described in Johnston et al. [49], placed in the middle of the chamber to avoid particle loss around the edges of the chamber. The air quality monitors were named lab-1, lab-2, lab-3 and lab-4. The sensors are controlled by a Raspberry Pi to which they are connected via USB. The Raspberry Pi is powered by Power-over-Ethernet. The sensors are controlled using Python 3.6 libraries [58,59,60,61,62]. The Alphasense OPC-R1 logged data every 10 s while the four other models of sensor were set to log data every 1 s. The RH and the temperature were recorded by each monitor with a Sensirion SHT35 [63]. To avoid any potential impact of the enclosure, the air quality monitors were placed inside the chamber without their waterproof enclosure (see Figure 2b). Each air quality monitor contains two of each of the five models of sensors tested as shown in Figure 1b.

### 2.2. Reference Instruments

A DustTrak DRX 8533 Desktop (TSI Inc., Shoreview, MN, USA), an Optical Particle Sizer OPS 3330 (TSI Inc.), and an Aerasense Nanotracer (Oxility BV, Best, Netherlands) were used to provide reference measurement results to evaluate the performance of the sensors. These instruments were chosen because they can provide a time resolution of 10 s, comparable to the low-cost sensors.

The DustTrak DRX 8533 Desktop was used as a reference instrument for PM_2.5_ concentration. It is an optical instrument based on 90${}^{\circ }$ light-scattering used by a number of laboratory studies [32,34,36,43,48,50]. It measures particles of diameter ≈0.1–15 μm and is calibrated both for size and particle mass using Arizona Road Dust (ISO 12103-1, A1). The DustTrak has been tested against gravimetric methods by previous studies [64,65]. Jayaratne et al. [32] observed that the DustTrak over-reports PM_2.5_ concentration for relative humidity >≈75%, in the absence of a heated inlet. To avoid the impact of humidity on the readings of the DustTrak, in this experiment, the DustTrak was fitted with a heated inlet which was activated for relative humidity >≈50%, measured by an external probe connected to the DustTrak. Rivas et al. [66] also identified technical problems affecting the performances of the DustTrak: an artefact jump in PM concentrations which can be corrected by the use of a zero calibration module and reporting of negative or concentration of 0 μg/m${}^{3}$ for low concentrations. The zero calibration module requires sampling intervals to be >2 min and as such was not used during this study but the artefact jump was also not observed during the study. However, during the experiments, the DustTrak reported some concentrations ≤0 μg/m${}^{3}$, these negative values were replaced by zero.

The OPS 3330 was used as a reference instrument for particle numbers between 0.3–10 μm divided into 16 size bins. The OPS 3330 is calibrated for size using Polystyrene Latex spheres which have a refractive index of 1.59 and has a flow rate of 1 L/min. It uses a laser at 660 nm and collects the 90${}^{\circ }$ ± 60${}^{\circ }$ scattered light on a wide angle spherical mirror [67].

The Aerasense Nanotracer uses a diffusion electrical charging method for counting particles between 10–300 nm. It was used as a reference instrument to measure the particles below the cut-off size of the low-cost PM sensors. It was calibrated by the manufacturer with KNO${}_{3}$ polydisperse particles with a mean diameter of 49.6
nm at 258,000 particles/cm${}^{3}$. Its airflow is 0.3–0.4 L/min.

The three reference instruments were set to take a reading every 10 s. The time constant of the DustTrak was set to 1 s to enable real time monitoring of rapid variation in PM during the course of the experiments.

### 2.3. Experimental Set-up

A total of 40 sensors were tested (eight units of each of five models from different manufacturers). The experiments took place inside a 1 m${}^{3}$ chamber constructed from Perspex and stainless steel held within aluminium frames. The air within the chamber was mixed with fans. The chamber was contained within a walk-in, temperature controlled room (Viessmann A/S), with the temperature held between 25.9
${}^{\circ }C$ and 28.7
${}^{\circ }C$. RH inside the test chamber was modulated between 55% and 90%, depending on the test conducted, through the introduction of filtered, dry air, or the same air humidified through a Nafion membrane using Mass Flow Controllers (MFCs) an MKS type 1179A 20 L/min and a type 1579a 100 L/min MFC. Chamber pressure was equilibrated prior to testing. A mist generator was also used to achieve elevated RH levels. PM was generated from incense sticks (Aphrodisia from Spiritual Sky) and introduced to the chamber via a 5 L/min flow of dry, filtered air through another type 1179A MFC. This equipment is summarised in Figure 2a. Candle generated PM was introduced via an angled chimney in the chamber wall (that was sealed when not in use) using a stearin candle from COOP Denmark A/S. RH and temperature inside the chamber were monitored via a Rotronic HygroPalm HP22 [68] and a DHT22 [69] controlled by a Raspberry Pi. PM was removed by filtering through an Electrostatic Precipitator (ESP), a model 50P (Expansion Electronics, Cartigliano, Italy).

### 2.4. Experiments Conducted

Two separate sets of experiments were conducted during this study. The first were tests under blank conditions during which the sensors were placed in the chamber, with all inlets sealed and the ESP activated to ensure zero air. This test lasted for 1.5
h. Appendix B
Figure A3 presents the measurements taken from the OPS and the DustTrak when the ESP was activated and shows the effective removal of particles by the ESP. For the second set of experiments, the sensors were placed in the chamber, initially at 23 ${}^{\circ }C$. Five different subsets of experiments were conducted at different RH. For each subset, the sensors were submitted to (1) at least six peaks of candle smoke; (2) one stable concentration of candle smoke; (3) at least six peaks of incense smoke; (4) one stable concentration of incense smoke. In this study, peaks of PM are defined as rapid variation over a short period of time of the order of 60 s. The profiles of candle smoke were generated by lighting and extinguishing a candle repeatedly under the chamber chimney until the targeted PM_2.5_ concentration, read on the DustTrak, was reached. This process produced candle smoke in a smouldering mode. According to Fine et al. [70], this mode produces mostly particles between 0.1–1.8 μm of diameter with particles size peaks between 0.4–0.8 μm. The incense concentrations were generated by lighting an incense stick just before turning on the MFC. After a few seconds, when the target concentration was reached, the MFC was turned off. According to Li and Hopke [71], incense smoke is composed of particles of diameters between 0.05–0.7 μm with a peak at 0.2
μm and it presents a lower hygroscopic growth ratio than candle smoke. The targeted amplitude of the peaks for both sources was between 20–50 $\mu g/m^{3}$ to mirror typical outdoor pollution conditions. Given the way the peaks were generated it was not possible to have a more precise target. After 10–20 s, the ESP was turned on to remove the particles until the mass concentration reported by the DustTrak was <5 μg/m${}^{3}$. The total duration of the peaks was around 1 to allow the DustTrak and the OPS, sampling every 10 s, to take sufficient measurements during the peaks. It would be equally interesting to study peaks of shorter duration but this was not possible in this study because of the time averaging used for the DusTrak and the OPS. A longer duration for the peaks was excluded because as time increases, the chances of detecting specific events of pollution decreases. The next profile was generated after 10–20 s.

During the tests, the RH was maintained around the targeted level by manually triggering the mister and for each experiments, the RH measured by the DHT22 had a range <2% (see Appendix B
Figure A4). The median RH recorded by the four SHT35 is used for the analysis in the paper as the diffusion of the water content is fast enough for the conditions in the direct vicinity of the sensor to impact the particles measured. Further details on the reasons for selecting this median are given in Appendix B
Section A.3.

RH was increased during the different experiments and the temperature was set at the beginning of the experiment but not controlled. The five sets of experiments were conducted at different RH levels, the recorded values for each experiment are shown in Table 2.

A number of sensors did not perform correctly during the experiments and were discarded from the analysis: one Honeywell HPMA115S0 reported concentrations >40 μg/m${}^{3}$ during the blank test; a second Honeywell HPMA115S0 reported concentrations with an offset of ≈40 μg/m${}^{3}$; one Alphasense OPC-R1 stopped reporting data after experiment one; two Plantower PMS5003 did not record data during the five experiments.

### 2.5. Data Analysis

The data analysis was conducted using R 3.5.1 (R Foundation for Statistical Computing, Vienna, Austria) and Prism 8 (GraphPad Software, San Diego, CA, USA) and the R scripts used to produce the results presented in this study are available http://doi.org/10.5281/zenodo.3695854. To compare the readings from the different instruments, an averaging period of 10 s was used and the readings were aligned to every 10 s. The underlying dataset for this study is available at Appendix A (http://doi.org/10.5281/zenodo.3695827). It contains the data measured by the low-cost PM sensors, by the temperature and relative humidity sensors and by the TSI DustTrak, the TSI OPS and the Aerasense Nanotracer.

#### 2.5.1. Lower Limit of Detection

There is no standardised method for determining the Lower Limit of Detection (LLOD) and the selection of a method can alter the results [73]. Three different methods have been compared. The first method defines the LLOD of the sensors as three times the standard deviation under blank conditions [74,75]. The second method, developed by Wallace et al. [65], considers the ratio of the mean and of the standard deviation of the multiple sensors of each model. The LLOD at 99% confidence level is defined as the concentration for which the ratio exceeds three. The third method [36,50,76], generally used for in-field determination of the LLOD, also requires the comparison with a reference instrument and defines the LLOD as three times the standard deviation divided by the slope of a linear model between the sensor and a reference instrument.

#### 2.5.2. Coefficient of Variation

The coefficient of variation or relative standard deviation is a measure of the precision of the measurement of the different models of sensors. It is calculated following NIOHS [77] guidelines. For each model of sensors, each unit is considered as a sample and the relative standard deviation between the units is calculated for every data point (i.e., one every 10 s):(1)$$CV_{t,model}=\frac{\sigma_{t,model}}{\mu_{t,model}}$$
where $\sigma_{t,model}$ and $\mu_{t,model}$ are respectively the standard deviation and the mean PM_0.1_ concentrations of the units of the model of sensor at the instant *t*. The coefficient of variation is then calculated as the pooled estimate of the relative standard deviation previously calculated:(2)$$CV_{model}=\sqrt{\sum_{i=1}^{k}\frac{\left(n_{i}-1\right)*CV_{i,model}^{2}}{\sum_{l=1}^{k}\left(n_{l}-1\right)}}$$
where *n*${}_{i}$ is the number of units used to calculate *CV*${}_{i,model}$ and *k* is the number of *CV*${}_{t,model}$ calculated for each model of sensor. A coefficient of variation of zero corresponds to a perfect precision and the US Environmental Protection Agency requires a coefficient of variation ≤0.1 for PM monitoring [78].

#### 2.5.3. Correction of the Delay

A short but significant delay was noticed between the peaks registered by two models of sensors compared to the DustTrak; the Honeywell HPMA115S0 and the Novafitness SDS018. The delay varies between units and between different experimental conditions. Potential explanations for this delay are discussed in Section 4. The value of the delay has been evaluated by applying a time lag to each individual time series and calculating the *R*${}^{2}$ of the linear model between the DustTrak and the resulting time series. The delay is then determined by the lag for which the *R*${}^{2}$ is maximum. Time lags varying between 1–100 s were observed for the sensors. The presence of this delay means that the results must be corrected before they can be compared, unavoidably introducing an amount of uncertainty. To correct this delay, Derivative Dynamic Time Warping (DDTW) [79] is used. Dynamic Time Warping (DTW) is a method used to align two time series that have similar shapes but are not aligned with each other, it computes a cost matrix between the different points of the two time series and then chooses an optimal path, based on a step pattern, in the matrix to match the time series. Different versions of DTW have been tested, with different step patterns and the method that DTW enables aligning the time series while not deforming the amplitudes of the sensors’ readings. The step pattern chosen is “mori2006” [80], the method is implemented using the R package ‘dtw’ [81].

#### 2.5.4. Influence of Humidity

The effect of humidity on the readings of the different sensors is evaluated on the data corrected for the delay. For each sensor, a linear model between the PM_2.5_ mass concentrations reported by the sensors and the DustTrak was fitted (x = DustTrak, y = sensors), for each experiment and for each of the two sources of PM. These linear models give the *R*${}^{2}$ coefficient which is an indication of the linear relationship between the two variables tested, and the slopes. Slopes with a *p*-value > 0.05 are rejected. Appendix B
Figure A5, Figure A6, Figure A7, Figure A8 and Figure A9 present the linear models obtained.

## 3. Results

This section presents the results obtained for: the LLOD, the time series of the experiments, the delay observed for some models of sensors, the coefficient of variation of the different models of sensors, the influence of humidity on the *R*${}^{2}$ coefficient and the slopes of each individual sensors and a comparison of the amplitude of the peaks registered before and after calibration.

### 3.1. Lower Limit of Detection

Figure 3 presents the LLOD of the different models of sensors calculated under blank conditions. The Alphasense OPC-R1 and the Sensirion SPS030 display similar LLOD, between 0.3–0.4 $\mu g/m^{3}$, with limited inter-unit variability. The Novafitness SDS018 also presents a limit of detection <0.4
μg/m${}^{3}$ but with a larger range of values. The Honeywell HPMA115S0 presents a LLOD with more inter-unit variability ranging from 0.3
μg/m${}^{3}$ to 1.2
μg/m${}^{3}$. The Plantower PMS5003 showed a LLOD of 0 μg/m${}^{3}$. Three models of sensors also report number concentrations and Table 3 reports the LLOD obtained by these sensors for the total number of particles reported. The values obtained by the Alphasense OPC-R1 and the Sensirion SPS030 are again similar but the values reported by the Plantower PMS5003 are several orders of magnitude lower. According to the manufacturer, the Plantower PMS5003 reports particle number concentration in $\#/100\;cm^{3}$ while the two other sensors report particle number concentration in $\#/cm^{3}$. During this experiment, the OPS reported a total particle number concentration of 40.2 #/cm${}^{3}$. The results from the Wallace et al. [65] method for the LLOD are given in Appendix B
Table A2, Table A3, Table A4, Table A5 and Table A6. All the models of sensors obtained a LLOD <1 μg/m${}^{3}$.

The third method for the LLOD requires the slope of the calibration of the sensors with a reference instrument. The values of the slopes obtained in this study, and presented later on in this paper, are variable, depending mainly but not exclusively on the source of PM. This method produced limits of detection varying widely across the experiments conducted.

### 3.2. Time Series of the Experiment

Figure 4 presents the time series of the first experiment conducted. Appendix B
Figure A10, Figure A11, Figure A12 and Figure A13 present the time series of the other four experiments. These figures show that the influence of the position of the air quality boxes on the readings has a lesser impact than the intra-model variability. They also show that all the peaks generated during the experiments have been recorded by the sensors although with different intensity and, for some models of sensors, with a delay, more pronounced for the later experiments, which is addressed in Section 3.3. The OPS bin size distribution reveals different size distributions for incense and candle smoke. The OPS shows that incense smoke contains proportionately more particles of diameter <0.374
μm than candle smoke which contains proportionately more particles of diameter >0.579
μm.

For incense, all the experiments had similar proportions of particles between 0.3
μm and 0.465
μm. Experiment 1 had the highest proportion of particles between 0.465–0.897 μm. Experiment 3 had the highest proportion of particles between 0.897–2.156 μm. Experiment 4 was mainly composed of particles <0.465
μm while presenting the highest total number of particles. Experiments 1 and 3 had 15% to 21% less total number of particles than Experiments 2, 4 and 5. Experiments 3 and 4 had the highest number concentrations of ultrafine particles. The peaks of incense range between 9–63 $\mu g/m^{3}$ with median concentration of 27 μg/m${}^{3}$ and a duration varying between 50–140 s.

For peaks generated by the candle, all the experiments had a similar distribution profile. The total number of particles during experiment 4 is 27% to 41% lower than during the other experiments while having by far the highest concentration of ultrafine particles. The number concentration of ultrafine particles was lower for experiment 3 and 5. The peaks of candle range between 9–100 $\mu g/m^{3}$ with median concentration of 34 μg/m${}^{3}$ and a duration varying between 50–100 s. Further details on: the size distribution, measured by the OPS; the characteristics of the peaks generated, the statistical differences between the peaks generated; and the number concentration of ultrafine particles measured by the Nanotracer; are available in Appendix B
Figure A1 and Figure A2 and Appendix B
Table A1. No significant differences were observed between the peaks generated during the different experiments.

### 3.3. Delay

A delay between the signal and sensor response registered by the Honeywell HPMA115S0 and Novafitness SDS018 sensors and the peaks registered by the DustTrak can be seen on Appendix B
Figure 4 and Figure A10, Figure A11, Figure A12 and Figure A13. The delay is variable through time and it increased as the experiments progressed: the delay was maximum during Experiment 5 which was the experiment the further away in time from the start. This delay is not linked to RH as it was observed to equally increase with time in another set of experiments not presented here which started at high RH and finished at low RH. The presence of this delay makes it impossible to compute the linear regressions required for the rest of the analysis without applying a correction. Figure 5 shows the delay calculated for the different models of sensors. The delay for the Honeywell HPMA115S0 sensors increased with time, starting at 5 s during experiment 1 and ranging between 15–75 s across the six operational sensors of this model. The delay for the Novafitness SDS018 increased with time like for the Honeywell HPMA115S0 but with much more inter-unit variability. The delay ranged from 14–17 s in experiment 1 and between 0–96 s overall. The Alphasense OPC-R1, the Plantower PMS5003 and the Sensirion SPS030 do not appeared to exhibit a delay when compared to the DustTrak.

The delay observed had to be corrected to enable the comparison with the reference instruments present. Simply shifting the data was not possible because: (1) the sampling period of the sensors and the reference instruments were not synchronised and a 10 s averaging and alignment had to be implemented prior to the analysis; (2) this implies an uncertainty of up to 10 s on the value of the delay given above.

The delay was corrected using DDTW, described in the Section 2 of this paper. Appendix B
Figure A14 presents results of the data correction method used and reveals that the method chosen enables shifting of the data while preserving the values of the original signal. The data corrected was used to conduct the analysis on the correlation.

### 3.4. Coefficient of Variation

Table 4 presents the coefficient of variation obtained for the different models of sensors for the data not delay corrected depending on the source of PM_2.5_ and the variation of the concentrations. Appendix B
Figure A15 present the same results but sorted by experiment. All of the sensor models showed a higher coefficient of variation for peaks of incense than for stable concentrations of incense. For candle, the Sensirion SPS030 and the Plantower PMS5003 obtained similar coefficients for both peak and stable concentrations while the three other models of sensors showed lower coefficients for stable than for peak concentrations. The Alphasense OPC-R1 is the model of sensor that presented the highest coefficients of variation for incense, both peaks and stable. The other sensors obtained lower results for incense than for candle. The Sensirion SPS030 and the Plantower PMS5003 both presented similar results for peaks and stable concentrations of candle. The Sensirion SPS030 obtained the lowest coefficients of variation of all models of sensor, with similar results for incense and candle peaks.

It is to be noted that the Honeywell HPMA115S0 and the Novafitness SDS018 showed better coefficient of variation on the delay corrected dataset, as presented in Appendix B
Table A7, with similar trends regarding the source and the rate of variation of the PM. The effect of the correction is more pronounced for the Novafitness SDS018, likely due to the fact that for this model the delay varied significantly between the different units.

### 3.5. Influence of Humidity

Figure 6 shows the *R*${}^{2}$ obtained by each individual sensor when fitting a linear regression model between the sensor and the DustTrak. All the models of sensor present a decrease in correlation for the fourth experiment for candle smoke. For candle smoke, the *R*${}^{2}$ decreases with increasing humidity from 68% RH but drops more for 76% RH than for 72–79% RH. For incense smoke, the effect of humidity appears to be less pronounced. The interquartile ranges for RH varying between 72–79% overlap for the Plantower PMS5003 and Sensirion SPS030.

The Plantower PMS5003 is the sensor that obtained the highest values for the two categories. The Sensirion SPS030 showed more intra-model variability for incense than for candle but with similar coefficients. The Honeywell HPMA115S0 and the Novafitness SDS018 present a drop for 76% RH for the incense PM while the Plantower PMS5003 and the Sensirion SPS030 present an increase for that experiment. The Alphasense OPC-R1 show low *R*${}^{2}$ for the incense PM with a considerable intra-model variability. The time series of the Alphasense OPC-R1 (Figure 4 and Appendix B
Figure A10, Figure A11, Figure A12 and Figure A13) show that this model of sensor did not correctly measure incense smoke during the five experiments. This model of sensor has stronger correlation for candle smoke. For all the sensors, the *R*${}^{2}$ dropped for candle on experiment 4 at 76% RH.

Figure 7 shows the slopes obtained for the five different experiments for the different models of sensor. The slope is an indication of the deviation of the sensors compared to the DustTrak: below one, the sensors under-report and above one, the sensors over-report. Appendix B
Figure A5, Figure A6, Figure A7, Figure A8 and Figure A9 present the plots of the linear model between each sensor and the DustTrak for each experiment conducted.

The Plantower PMS5003 obtained higher slopes for incense than for candles generated PM. The slopes for candles PM increase from experiment 3 at 72% RH meaning that the Plantower PMS5003 under reports less for higher humidity compared to lower humidity. The same effect can be observed for the PM generated by incense with a trend similar to the *R*${}^{2}$ but in the opposite direction. For incense PM, the Plantower PMS5003 presents an increase for experiment 3. In contrast, the response of the Sensirion SPS030 and the Novafitness SDS018 drop for this experiment. The effect is opposite for the other sensors, where the slope decreases for RH greater than 72%. The Sensirion SPS030 shows similar slopes for incense smoke and candle smoke. The Honeywell HPMA115S0 showed very similar slopes between peaks and stable concentrations of both sources of PM but the slope varied significantly between the two types of sources with slopes close to 1 for candle and slopes below 0.5 for incense. The Novafitness SDS018 also showed similar slopes for peaks and stable concentrations of both sources of PM but with more similar slopes across the two sources of PM. The Alphasense OPC-R1 showed very different slopes for the incense and the candles with very low slopes for incense smoke. It over reported candle particles. The different units of this model have a wide range of slopes. The Plantower PMS5003 and the Honeywell HPMA115S0 are the models presenting lower intra-model variability.

### 3.6. Amplitude of the Peaks

Following the results obtained from the linear models for each source and experiment, each individual sensor has been corrected as follows:(3)$$PM_{2.5,corrected}=\frac{PM_{2.5,measured}-intercept}{slope}$$

The maximum value of the peaks recorded by each sensor have been manually extracted and compared to the values of the peaks measured by the DustTrak. Figure 8 presents the range of the ratio obtained between the peaks of the sensors and the DustTrak. A ratio <1 means that the sensor under reported and a ratio >1 that the sensor over reported compared to the DustTrak. For the Plantower PMS5003, the calibration lead to ratios close to one in all cases. The same is true to a lesser extent for the Sensirion SPS030 but with a more pronounced increase of the range of ratios observed. For the Honeywell HPMA115S0, the calibration for the candle smoke moved the ratio further away from one but for the incense smoke, it moved the ratios closer to 1. For the Novafitness SDS018 the ratios were closer to one for both sources of PM but with an wider range of values for incense generated PM. The Alphasense OPC-R1 is the only model of sensor that showed a ratio greater than one. For candle particles, the calibration brought the ratios closer to one with a range reduction while for incense particles, the ratios were also brought closer to one but with a wide range of values.

## 4. Discussion

In this study, the lower limit of detection of the sensors has been determined using three different methods, the first based on the standard deviation of the sensors under blank conditions, the second on the evolution of the ratio between the mean and the standard deviation with increasing concentrations of PM and the third based on the standard deviation of the sensors at low concentrations after calibration. A series of five experiments were conducted for two candle and incense generated PM at five different levels of RH. The time-series of the experiments were presented and analysed. Then for some model of sensors, the time-series were adjusted to compensate the delay observed. The influence of humidity on the slope and the Pearson coefficient of the linear model between the sensors and the DustTrak was analysed. Finally, the performance of the calibration of the sensors through the linear model on the values of the peaks reported by the sensors was assessed. While providing useful insight on the performances of the sensors, this study has some limitations that must be taken into account for extrapolating its results. It only tested the sensors against two sources of pollution that are related to combustion. Environmental particles, in particular from non combustion sources, will have different physical and chemical properties including their hygroscopicity, refractive index and particle size. The range of temperature used did not reflect the range of temperatures experienced in outdoor condition and the range of RH was limited. Finally, the reference instruments used to measure PM did not measure PM mass directly.

The first two methods employed for determining the LLOD of the sensors produced similar results and are similar to the limit of detection measured by Northcross et al. [74] and Austin et al. [46] on different models of light-scattering PM sensors during laboratory studies. The LLODs reported by outdoor studies are generally higher than the ones reported here: Sayahi et al. [50] applied the third method and found a LLOD of 6 μg/m${}^{3}$ for the Plantower PMS1003 and PMS5003; Zikova et al. [82] applied the second method and obtained a LLOD of 10 μg/m${}^{3}$ for Speck monitors. The objective of determining the LLOD for low-cost PM sensors is to be able to determine the value above which they can reliably measure PM. The results of this study suggest that the LLOD determined through laboratory study may not be sufficient to evaluate the accuracy and precision of these sensors at low concentrations of PM. The LLOD is difficult to evaluate outdoor because most of the reference instruments available only report hourly data making the comparison to the high frequency data produced by the sensors impossible and most studies only determine at best the LLOD for hourly data. This shows the need to have reliable instruments that measure and report data at higher frequency than what is currently available.

To the best of our knowledge, this study is the first to assess the delay in the response of low-cost sensors relative to peaks of pollution. A delay was observed on two out of the five models tested. Depending on the models of sensor, the delay varied between a few seconds to ≈2 min. The Honeywell HPMA115S0 and the Novafitness SDS018 are the only models examined here that only report PM mass concentration and no particle count which may suggest they use a different method for inferring the number of particles from the electrical signal generated by their photodetector. The Honeywell HPMA115S0 applies a 10 s moving average to its raw readings. However, these elements alone cannot fully account for the delay observed. The Novafitness SDS018 also present a high inter-unit variability for the delay. The existence of this delay implies that these sensors are nopt suitable to be used when a time averaging period <2 min is required. The delay correction method proposed here requires a post-treatment of the data with a reference instrument and is not suitable for real-time correction but can still address the issue of time alignment when multiple instruments are used during an experiment. It should only be considered if the delay corrected sensors present better performances than other sensors. For applications that require a high time granularity, it is recommended to avoid sensors presenting that exhibit a delay in response to transient events.

The different response of low-cost PM sensors to sources of pollution is a known and well studied issue [28] and the goal of this study was not to characterise this response. The two different sources were used to acknowledge this effect but also to have an indication on the dependence of the performances of the sensors on the size distribution of the PM. The different performances of the sensors with candle and incense smoke may be partly attributed to the size distribution of the two smokes with the candle smoke containing larger particles than the incense smoke. The Plantower PMS5003 and the Novafitness SDS018 are the only two models of sensors that presented similar *R*${}^{2}$ for candle and incense particles while the three other models of sensors presented lower *R*${}^{2}$ for incense. The Plantower PMS5003 is the only model of sensor that showed a greater slope for incense than for candle suggesting that these sensors are best suited for measuring the smallest particles rather than bigger size fractions. This is further supported by the lower coefficients of variation obtained by this model of sensor for incense than for candle. Conversely, the similar *R*${}^{2}$ showed by this model of sensor for the two sources of pollution may suggests that this result from its factory calibration rather than from its actual capacity at differentiating size distribution of PM. The Sensirion SPS030 showed similar slopes for the candle and for the incense, potentially showing that this model of sensor can perform well for particles with different size distributions, which is also supported by the similar coefficients of variation obtained for the two types of smokes. The low signal shown by the Alphasense OPC-R1 for incense may be explained by the fact that most of the particles in the incense smoke are outside of its range of detection which is 0.4
μm. It would be important to study this sensor with an additional source of pollution with a higher mean diameter.

The variation between experiment 3 and experiment 4 for incense PM may be due to the size distribution, where experiment 4 had mostly ultrafine particles and experiment 3 had more of the larger particles than any other experiments with similarly high levels of ultrafine particles than experiment 4. The particle size distributions of experiments 3 and 4 were different from the particle size distributions of 1, 2 and 5 which were more similar to each other. This supports the observation that the Honeywell HPMA115S0 and Novafitness SDS018 seem to perform better for larger particles and the Plantower PMS5003 and Sensirion SPS030 better at smaller particles. This may also suggest that the sensors are more susceptible to changes in the particle size distribution than to relative humidity.

The drop in the slope and Pearson coefficient for candle particles seen during experiment 4 can be explained by a greater proportion of small and ultrafine particles than in the other experiments. This experiment showed the highest number of ultrafine particles and the lowest total number of particles >0.3
μm. The low-cost sensors studied here cannot detect particles <0.3
μm while the DustTrak can detect particles down to 0.1
μm. The two different smokes also have different refractive indices and hygroscopic properties which will influence the readings of the sensors [70,71]. The hygroscopicity will also affect directly how the particles react with varying humidity. After taking into account the particle distributions of experiments 3 and 4, it can be observed that the Plantower PMS5003 tends to have higher slopes for RH above 72% while the four other models of sensors the slope decreases above 72%. The range of RH attained in this study may not be sufficient to capture the impact of this environmental factor and the behaviour of the sensors may change for higher levels of RH. While it is possible to observe the impact of RH on the readings of the sensors, its effect is limited and is certainly less pronounced than the size distribution of the particles and the source of pollution although it is possible that using different sources of pollution would yield different results. Further analysis focusing on the size distribution reported by the Plantower PMS5003, the Sensirion SPS030 and the Alphasense OPC-R1 rather than on the mass concentration, may yield more details on the capacity of these models of sensors to track sources of pollution.

All the sensors registered the different peaks of pollution generated, with varying intensity depending on the model considered. The coefficient of variation showed that the sensors were generally more precise when measuring stable concentrations of pollution than peaks of pollution except for the Plantower PMS5003 and the Sensirion SPS030 for candle. The Sensirion SPS030 yielded coefficients of variation of 0.19 for both incense and candle peaks showing that while they did not meet the 0.1 requirement for legal monitoring of PM the two models of sensors still have the capacity to qualitatively monitor peaks of pollution. The Plantower PMS5003 is well-able to characterise peaks of incense smoke, but less able for candle particles. After calibration, the Plantower PMS5003 is the model of sensor that yielded the best results for the ratio of peaks of PM for both sources of PM, followed by the Sensirion SPS030. This illustrate the need to calibrate the sensors in conditions as close as possible to their actual condition of use. Given, the different sensitivity of the model of sensors to different sources of pollution and of different size distribution, it may be fruitful to use a combination of different models to track sources of pollution. A combination of two sensors may also prove interesting for the post-processing of the data to detect faults or erroneous data by comparing the two models of sensors. Johnston et al. [49] present the evolution the versions of an Air Quality Internet of Things device from hosting four sensors to ten sensors going from $900 to $1000. Camprodon et al. [83] present another air quality monitoring device which can host one or two Plantower PMS5003 for respectively $120 or $170. Other commercially available devices are using two PM sensors [84,85]. The above examples show that it is technically and economically possible to use two different models of low-cost PM sensors at the same time. Given the potential of providing us information beyond PM mass concentrations, it is advised to further test this solution in field conditions with sensors deployed for a long term as a network of sensors.

## 5. Conclusions

This study determined the LLOD of the sensors using two different methods, both of which showed a LLOD <1 μg/m${}^{3}$. However, it is argued that these methods do not give a full characterisation of the precision and accuracy of the sensors when sampling low concentrations of PM. Indeed, the results obtained by these two methods are an order of magnitude different from the values obtained by other outdoor studies. The delay observed on some models of sensors has implications regarding the choice of sensors to be used for measurements with a sampling period <1 min. The correction method used here, while being suitable for post-treatment of the measurement, would not be suitable for real time correction unless the sensors were collocated with a reference instrument. The coefficient of variation calculated for the sensors are above the maximum required for ambient PM monitoring but they showed promising results regarding the use of some of the sensors tested to identify transient pollution events and track events of pollution if placed as a network. This coefficient enabled differentiation of the performance of the sensors, for sources of pollution and for peaks against stable concentration of pollution, and is a valid metric for the cross-comparison of the sensors.

The sensors tested were able to detect the <1 min events of pollution generated from both sources, although with varying degrees of precision. This suggests that the short time and intensity peaks observed in the time-series of the sensors during outdoor deployment study are genuine.

The sensors showed a strong sensitivity to the source of PM and to the size distribution of the particles generated. This sensitivity was different for different models of sensors suggesting that using a combination of different models of sensors could be interesting to provide additional data to PM mass concentration and to assist fault detection and data post-processing. Further work is required to understand the capacity of the sensors to differentiate particle sizes and would benefit from a detailed study of the particle size distributions reported by the sensors.

The calibration performed between the sensors and the DustTrak reduced the bias between the sensors and the DustTrak for some of the sensors but not for all. The calibration also yielded different results for the two sources of pollution suggesting that some sensors may be better suited than others to track specific sources of pollution, independent of the aerosol source that has been used for their initial factory calibration.

The performance of some of the models of sensors for peaks of pollution, their different sensitivity to sources and particle distribution, and their capacity to measure <10 s suggests that using a combination of PM sensors may improve the current capacity to identify transient pollution events and track sources or events of pollution. The deployment of a network of a combination of low-cost PM sensors around an urban area including collocations with reference grade instruments is the next step to achieve this goal.

## Figures and Tables

**Figure 1 sensors-20-02219-f001:**
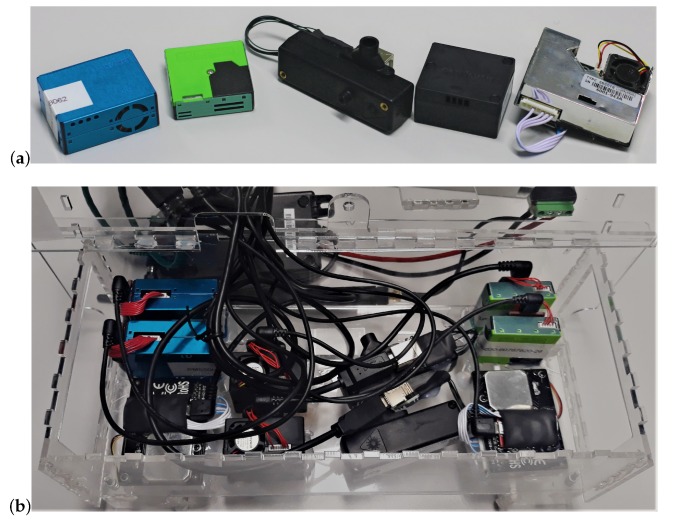
(**a**) Picture of the sensors tested from left to right: Plantower PMS5003, Sensirion SPS030, Alphasense OPC-R1, Honeywell HPMA115S0 and Novafitness SDS018. (**b**) Position of the sensors tested within each air quality monitor. From left to right, top to bottom: two Plantower PMS5003, one Novafitness SDS018, two Honeywell HPMA115S0, two Alphasense OPC-R1, two Sensirion SPS030 and one Novafitness SDS018. All the inlets are facing down.

**Figure 2 sensors-20-02219-f002:**
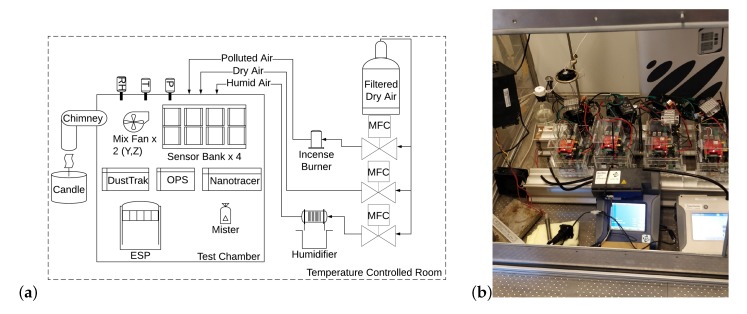
(**a**) Schematic showing the arrangement of the test chamber and supporting equipment. (**b**) Image showing the air quality boxes located in the test chamber.

**Figure 3 sensors-20-02219-f003:**
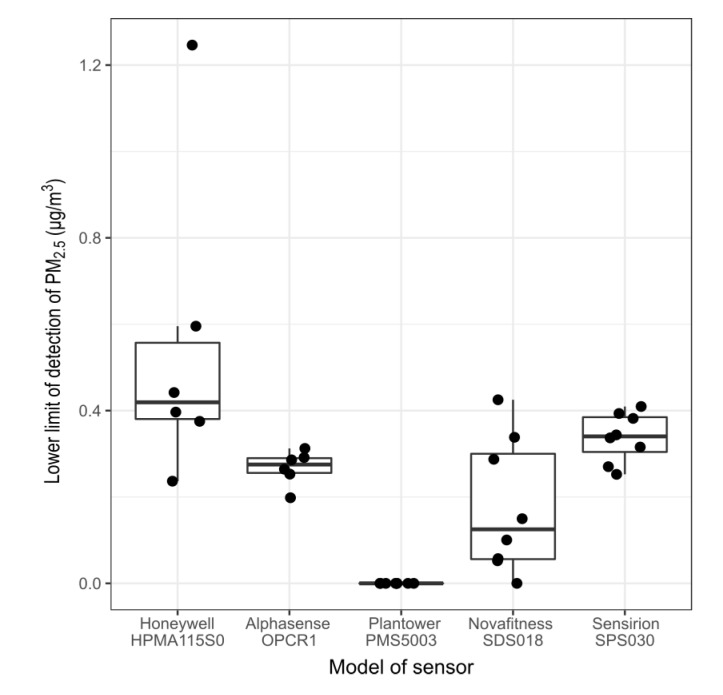
Lower limit of detection of the different models of sensors, under blank conditions, reported in PM_2.5_
$\mu g/m^{3}$. The box and whisker plot horizontal lines represent, from bottom to top, the lower quartile, the median and the upper quartile. The vertical lines are drawn to the smallest and the largest data point that fall within 1.5 times the interquartile range below the lower quartile and above the upper quartile respectively. Each dot represents the value obtained by individual sensors.

**Figure 4 sensors-20-02219-f004:**
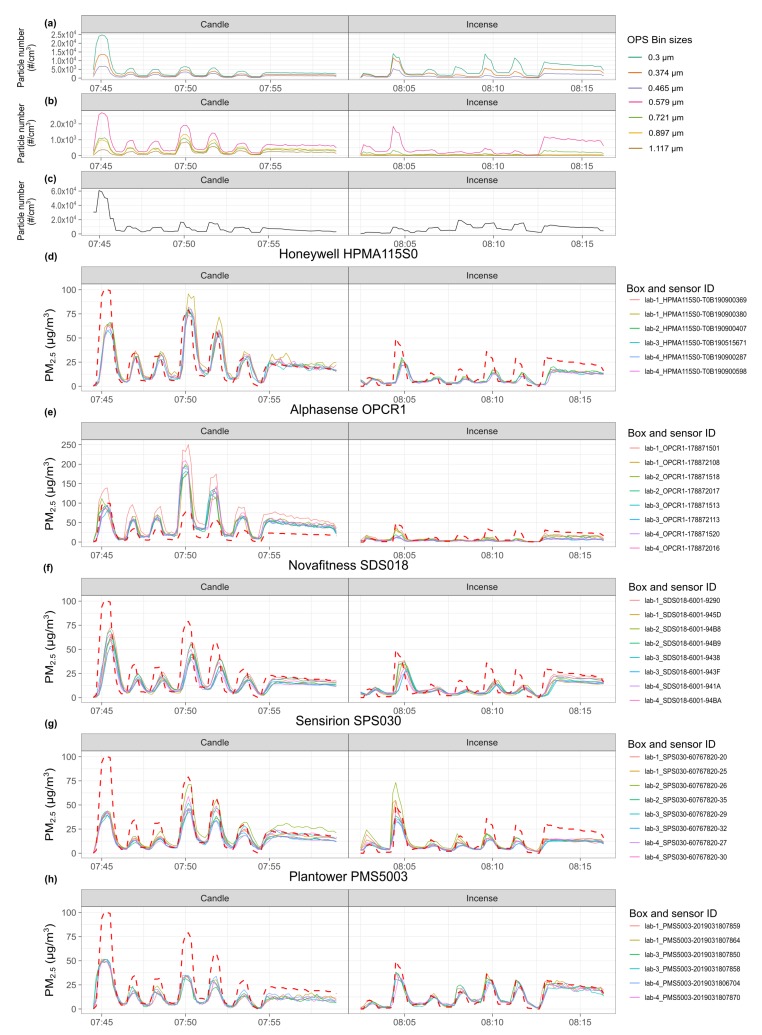
Time series of experiment 1. From top to bottom: (**a**,**b**) TSI 3330 OPS Bin size distribution; (**c**) Nanotracer ultrafine particles number concentration /mL; (**d**–**h**) PM_2.5_ of the sensors for each model of sensor, against DustTrak (red dotted line). The sensors are named according to the box they are placed in and their serial number.

**Figure 5 sensors-20-02219-f005:**
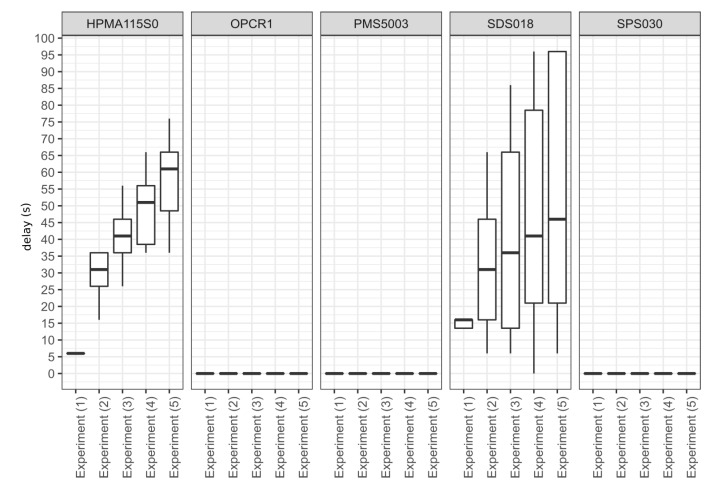
Comparison of the delay of the different model of sensors determined by calculating the maximum *R*${}^{2}$ obtained with the DustTrak by applying different time lag to the readings of each of the sensor tested, by model of sensor and by experiment conducted. The box and whisker plot horizontal lines represent, from bottom to top, the 1st quartile, the median and the 3rd quartile. The vertical lines represent the data points that fall within 1.5 times the interquartile range below the 1st quartile and above the 3rd quartile. Values outside these range are considered as outliers and plotted as points (none are present here).

**Figure 6 sensors-20-02219-f006:**
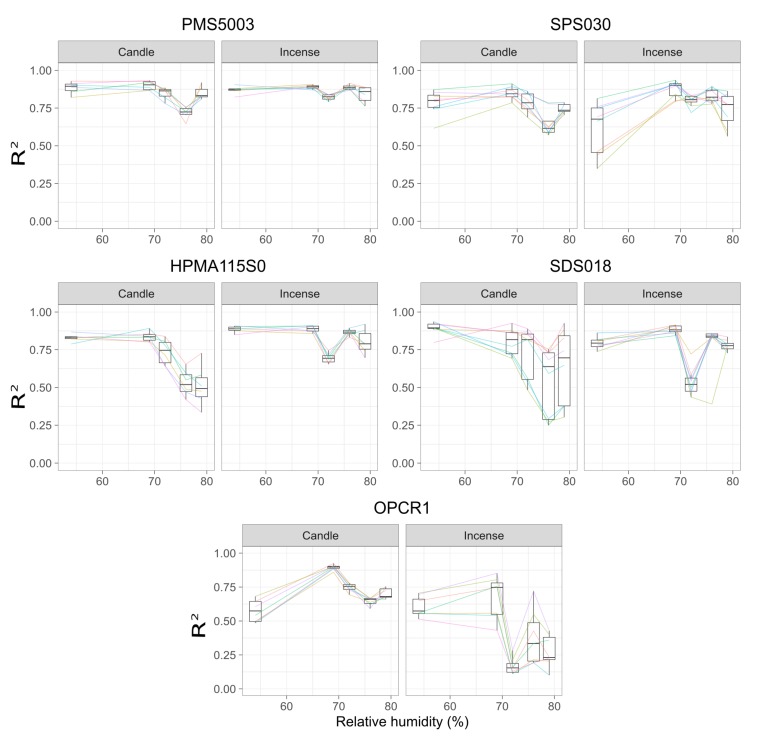
*R*${}^{2}$ of the linear regression between the readings of the sensors and the DustTrak for different level of relative humidity (each level of relative humidity corresponds to experiments 1 to 5), for candle and incense generated concentrations of PM_2.5_ (peaks and stable concentrations not differentiated). For each graph, the colours correspond to the sensors of the model tested. Only the points with a *p*-value < 0.05 have been considered. The box and whisker plot horizontal lines represent, from bottom to top, the lower quartile, the median and the upper quartile. The vertical lines are drawn to the smallest and the largest data point that fall within 1.5 times the interquartile range below the lower quartile and above the upper quartile respectively. Values outside these range are considered as outliers.

**Figure 7 sensors-20-02219-f007:**
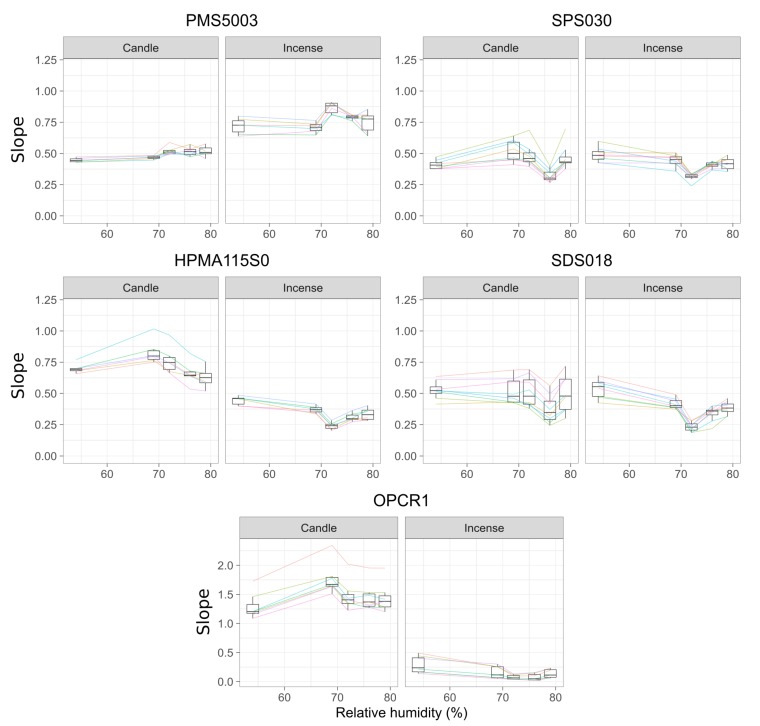
Slopes of the linear regression between the readings of the sensors and the DustTrak (y = sensor, x = DustTrak) for different level of relative humidity (each level of relative humidity corresponds to experiments 1 to 5), for candle and incense generated concentrations of PM_2.5_ (peaks and stable concentrations not differentiated). For each graph, the colours correspond to the sensors of the model tested. The colours are consistent with Figure 6. Only the points with a *p*-value < 0.05 have been considered. The box and whisker plot horizontal lines represent, from bottom to top, the lower quartile, the median and the upper quartile. The vertical lines are drawn to the smallest and the largest data point that fall within 1.5 times the interquartile range below the lower quartile and above the upper quartile respectively. Values outside these range are considered as outliers.

**Figure 8 sensors-20-02219-f008:**
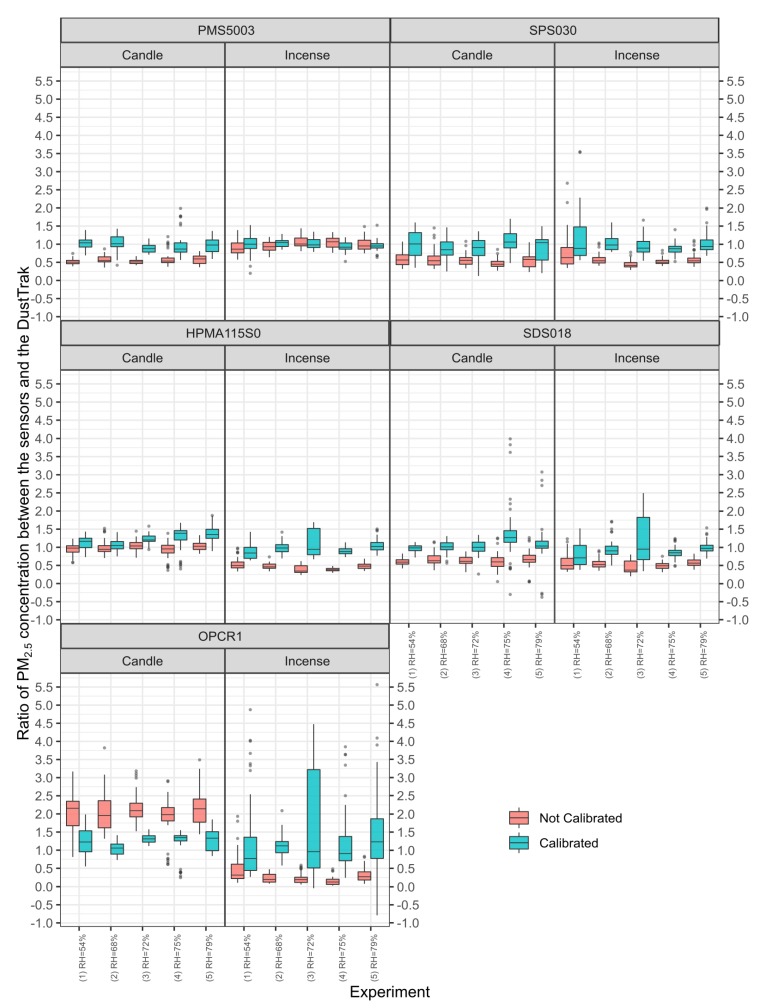
Ratio of the maximum value of the peaks measured by the sensors against the maximum value of the peaks measured by the DustTrak before and after calibration for each experiment. The box and whisker plot horizontal lines represent, from bottom to top, the lower quartile, the median and the upper quartile. The vertical lines are drawn to the smallest and the largest data point that fall within 1.5 times the interquartile range below the lower quartile and above the upper quartile respectively. Values outside these range are considered as outliers and plotted as points.

**Table 1 sensors-20-02219-t001:** Manufacturer specifications of the five models of low-cost particulate matter (PM) sensors tested.

Sensor	Particle Range (μm)	Concentration Range (μm${}^{3}$)	Response Time (s)	Maximum Humidity (%)	Cost (Euro)	Size (mm) (H × W × D)
Honeywell HPMA115S0	-	0–1000	<6	95	35	36 × 29 × 23
Alphasense OPC-R1	0.4–12.4	0.01–1,500,000	1–30	95	123	26 × 36 × 72
Novafitness SDS018	>0.3	0.0–999.9	<10	90	20	59 × 45 × 20
Sensirion SPS030	0.3–10	0–1000	1	95	42	41 × 41 × 12
Plantower PMS5003	0.3–10	1–1000	10	99	15	50 × 38 × 21

**Table 2 sensors-20-02219-t002:** Median Humidity and Temperature readings averaged from the four Sensirion SHT35, for each of the five experiments performed. The values are reported ±2.5 Median Absolute Deviation [72].

Experiment Number	Relative Humidity (%)	Temperature (${}^{\circ }C$)
1	54±3	25.6±2.2
2	69±2	28.2±1.1
3	72±2	28.4±1.1
4	76±3	28.7±1.1
5	79±4	28.7±1.5

**Table 3 sensors-20-02219-t003:** Range of the lower limit of detection of Alphasense OPC-R1, Plantower PMS5003 and Sensirion SPS030, under blank condition, reported in number of particles per mL.

Sensor Model	Range of Lower Limit of Detection
	(#/cm${}^{3}$)
Alphasense OPC-R1	8.12–13.47
Plantower PMS5003	0.08–0.24
Sensirion SPS030	9.93–16.09

**Table 4 sensors-20-02219-t004:** Coefficient of variation obtained by the different models of sensors for different sources and different variations of PM sources where *n* is the number of unit of each model used for the calculation and *k* is the number of data points available.

Sensor Model	Number of Units (*n*)	Incense Peaks	Incense Stable	Candle Peaks	Candle Stable
Honeywell HPMA115S0	5	0.35 (*k* = 261)	0.27 (*k* = 136)	0.43 (*k* = 289)	0.32 (*k* = 130)
Plantower PMS5003	6	0.25 (*k* = 260)	0.14 (*k* = 137)	0.30 (*k* = 289)	0.31 (*k* = 130)
Novafitness SDS018	8	0.42 (*k* = 261)	0.35 (*k* = 137)	0.56 (*k* = 289)	0.47 (*k* = 130)
Sensirion SPS030	8	0.19 (*k* = 261)	0.12 (*k* = 137)	0.19 (*k* = 289)	0.20 (*k* = 130)
Alphasense OPC-R1	6	0.53 (*k* = 261)	0.48 (*k* = 137)	0.30 (*k* = 289)	0.24 (*k* = 130)

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
