# Peer review of "Laboratory Comparison of Low-Cost Particulate Matter Sensors to Measure Transient Events of Pollution"

_sensors, 2020, doi:10.3390/s20082219_

Round 1

Reviewer 1 Report

Dear Authors,

The article presents a significant amount of data and analysis for the shared study of different types of sensors, with the article having an adequate structure.

In the opinion of this reviewer, I request that:

- The scientific and technological contribution of the results of the article should be highlighted clearly and objectively

- A brief study is presented on the applicability and economic feasibility of implementing the using a combination of some of the different models to take advantage of the additional information made available by their differential response.

Finally, highlight the advantages obtained by using the proposed arrangement.

Contextualize the information in the conclusions of Link The underlying dataset is available at http://doi.org/10.5281/zenodo.3695827.

I suggest adapting the summary of the article because it is the same as that found in the Link: http://doi.org/10.5281/zenodo.3695827.

Success

Author Response

"Dear Authors,

The article presents a significant amount of data and analysis for the shared study of different types of sensors, with the article having an adequate structure.

We would like to thank the reviewer for this thorough review that helped to improve the quality of the paper and for acknowledging the amount of data and analysis it contains.

In the opinion of this reviewer, I request that:

- The scientific and technological contribution of the results of the article should be highlighted clearly and objectively

The 5 aims of the study have been clearly outlined at the end of the introduction. The conclusions have been better highlighted. The abstract has also been edited. We would like to thank the reviewer for this request that enabled us to better highlight the contribution and the conclusion of this paper.

- A brief study is presented on the applicability and economic feasibility of implementing the using a combination of some of the different models to take advantage of the additional information made available by their differential response.

We agree with the reviewer regarding the value of conducting such an analysis and some elements have been added in the last paragraph of the discussion regarding the applicability and economic feasibility of the method proposed. We would like to thank the reviewer for this comment that improved the assertiveness of the study. However, it is beyond the scope of the work outlined in the paper. Assessing various networks for various applications (if done well such as in Karigulian et al. (2019)) would not be a brief study. The paper is a necessary step towards others being able to undertake that work.

(1). Karagulian, F. et al. Review of sensors for air quality monitoring. JRC Technical Reports (2019). doi:10.2760/568261

- Finally, highlight the advantages obtained by using the proposed arrangement.

More details have been added to the last paragraph of the Discussion regarding the advantages obtained by using the proposed arrangement and they have been better highlighted in the Conclusion.

- Contextualize the information in the conclusions of Link The underlying dataset is available at http://doi.org/10.5281/zenodo.3695827.

The link to the dataset has been placed within “Supplementary Materials” subsection at the end of the conclusion.

- I suggest adapting the summary of the article because it is the same as that found in the Link:

The description of the dataset has been updated to remove the abstract of the paper. A link to the paper will be added to the description of the dataset when published.

Reviewer 2 Report

Very interesting paper and very interesting work. The work carried out is presented in a very thorough way, which in a way do not help for a fast reading but help the interested reader to fully understand the work. The author also gave the main criticism on its own work in the discussion. Congratulations to the author for this work.

Author Response

We would like to thank the reviewer for this positive review.

Reviewer 3 Report

The manuscript reports on the comparison of a few low-cost sensor models with higher-grade devices and concludes a desirable monitoring performance can be achieved with these models. Overall, the manuscript is well-composed and to the point. The study is conducted appropriately with the claims and conclusion matching the presented data. The reviewer would suggest the article to be published in Sensors.

Minor comments:

1. What is the significance of measuring pollution spikes shorter than 1 minute? Why 1 min and not a shorter or longer duration?

2. Line 60: “… to enable citizens to make informed …”

3. Indicate panels (a) and (b) in Figures 1 and 2.

Author Response

The manuscript reports on the comparison of a few low-cost sensor models with higher-grade devices and concludes a desirable monitoring performance can be achieved with these models. Overall, the manuscript is well-composed and to the point. The study is conducted appropriately with the claims and conclusion matching the presented data. The reviewer would suggest the article to be published in Sensors.

We would like to thank the reviewer for their appreciation of the study.

Minor comments:

1.What is the significance of measuring pollution spikes shorter than 1 minute? Why 1 min and not a shorter or longer duration?

We would like to thank the reviewer for raising this point that required clarification in the paper. Some details have been added to section Methods, sub-section Experiments conducted at the end of the first paragraph.

The DustTrak and the OPS were set to read every 10s so to be able to take multiple measurements during a peak, the duration of the peak was about 1min. It would be maybe more interesting to study shorter peaks but it was not feasible with the equipment available during this study.

Not longer than 1 min because as time increases, the chances of detecting specific events decreases. 

2. Line 60: “… to enable citizens to make informed …”

These items have been amended. Thank you for noticing them.

Reviewer 4 Report

Unfortunately, I must write that my opinion is that the work presented in this article is not enough to support the publication. My general comment is that the quantity and quality of experiments and data interpretation are too limited to be useful and practical benefit for future readers.

The scope of the work is limited to reporting results from short-term durations co-locations. The sensor tests are for about 1,5 h on each of and experiencing a few conditions of PMs concentrations. Whilst the authors don’t claim they are doing more than this in this study,what is relevant for the practical deployment of these sensors is how they perform over extended periods (days/weeks) and multiple periods of use and no-use, and in environments where the variables change on relatively fast timescales.

The authors use DustTrak DRX 8533 as a reference, however the determination method used in this equipment is not the same as that recommended in the European standard EN 12341: 2014 for Ambient air. The standard method is the gravimetric measurement method for the determination of the PM10 or PM2,5 mass concentration of suspended particulate matter. This question must be justified and clarified.

What were the criteria used to choose the sensors?

As mentioned above, the experiments were short- term durations and it is suggested to be extended to other sources types. The particulate matter coming from the smoke of a candle or an incense stick are different from those from a rod traffic or from burning wood material (main sources in a city).

Is suggested in figure 2a), the indication of the air outlet. How is homogeneous air mixing guaranteed? How thus Electrostatic Precipitator (ESP) assure clean air inside the camera? This question must be justified and clarified.

Author Response

Unfortunately, I must write that my opinion is that the work presented in this article is not enough to support the publication. My general comment is that the quantity and quality of experiments and data interpretation are too limited to be useful and practical benefit for future readers.

The scope of the work is limited to reporting results from short-term durations co-locations. The sensor tests are for about 1,5 h on each of and experiencing a few conditions of PMs concentrations. Whilst the authors don’t claim they are doing more than this in this study,what is relevant for the practical deployment of these sensors is how they perform over extended periods (days/weeks) and multiple periods of use and no-use, and in environments where the variables change on relatively fast timescales.

We would like to thank the reviewer for this thorough review which helped improve the content of the paper. The limitations of the study are highlighted in the Discussion at the end of the first paragraph. The interpretation of the data and potential applicability have been detailed in the discussion. The full underlying dataset is published with the paper so it can be validated and extended if needed.

We agree that the long term performance of the sensors is entirely relevant, including periods of being used and not being used. And, as the reviewer points out, that was not the purpose of this study. In this study we focused on the response of the sensors at relatively fast timescales and carried out a series of tests in which we exposed the sensors to a rapid spike in concentration. We agree with the reviewer, that the response on relatively fast timescales is very relevant for practical deployment and have done our best to address the question with the experiments, analysis and discussion presented in the paper. The analysis of the response of these sensors to rapid spikes in concentration, conducted in this paper, as not been undertaken in the literature, to the best of our knowledge. One specific contribution we may note is the observation of a significant time delay in two of the models of sensors between pollution pulse and output signal.

Outdoor long term collocations of sensors at monitoring are necessary to the study of the long term performances of these sensors. In Bulot et al. (2019)1, we have analysed the performances of some of these sensors over a year long period. One of the goal of the current study is to generated spikes of pollution under controlled conditions to further investigate phenomena observed during the deployment.  Namely to assess how these sensors respond to short-time variation of PM.

Further details on potential applications, including a short cost analysis have been added to the last paragraph of the Discussion.

(1). Bulot, F. M. J. et al. Long-term field comparison of the performances of multiple low-cost particulate matter sensors in an urban area. Sci. Rep. 9, 1–16 (2019).

The authors use DustTrak DRX 8533 as a reference, however the determination method used in this equipment is not the same as that recommended in the European standard EN 12341: 2014 for Ambient air. The standard method is the gravimetric measurement method for the determination of the PM10 or PM2,5 mass concentration of suspended particulate matter. This question must be justified and clarified.

The DustTrak was evaluated against gravimetric methods in Matti Maricq(2013)1. Their conclusions are that the DustTrak model 8520 provides accurate PM data that correlate well with standard gravimetric methods when there is little variation of particle morphology and composition to the A1 test dust used for the factory calibration. Wallace et al. (2011)2 also drew similar conclusions. We thank the reviewer for raising the issue and have now added mention of the Maricq paper to the manuscript. While we agree with the reviewer, unfortunately, gravimetric methods are not suited for measuring the short-lived pollution events that are one of the focuses of this study. It is not possible to obtain the time-resolution needed using gravimetry. For example, the sampling period in the European standard is 24 hours, whereas in this study pollution spikes lasting <1 minute are investigated. Gravimetric instruments do not provide data at a sufficient resolution to compare with the sensors.

The DustTrak is widely used in air quality studies. A number of these previous studies are referenced in the manuscript (references 30, 32, 34, 41, 46, 48).

References:

(1). Maricq, M. M. Monitoring Motor Vehicle PM Emissions: An Evaluation of Three Portable Low-Cost Aerosol Instruments. Aerosol Sci. Technol. 2013, 47 (5), 564–573. https://doi.org/10.1080/02786826.2013.773394.

(2). Wallace, L. A. et al. Validation of continuous particle monitors for personal, indoor, and outdoor exposures. J. Expo. Sci. Environ. Epidemiol. 21, 49–64 (2011).

What were the criteria used to choose the sensors?

We included sensors popular in the scientific literature like the Plantower or the Alphasense, and some new sensors like the Sensirion or rarely tested sensors like the Honeywell and Novafitness.

As mentioned above, the experiments were short- term durations and it is suggested to be extended to other sources types. The particulate matter coming from the smoke of a candle or an incense stick are different from those from a rod traffic or from burning wood material (main sources in a city).

This is certainly true and thank you for the suggestion, we have included this in our suggestions for future work in the conclusions. The results we have are a first indication of the influence of particle composition on the response of these low-cost sensors.

Is suggested in figure 2a), the indication of the air outlet. How is homogeneous air mixing guaranteed? How thus Electrostatic Precipitator (ESP) assure clean air inside the camera? This question must be justified and clarified.

Thank you very much for this comment. It is an important aspect of the experiment that was not clear in the initial text. Some details have been added to the Methods section about the mixing of the air and the functioning of the ESP.

To clarify, the air outlets are sealed during the experiments, not open. We have revised the text to make this clear. A figure has been added to the annex (Supplementary Figure A3) to show the rapid and efficient cleaning of the chamber with the ESP, recorded by the reference instruments. These stable levels, over extended periods, demonstrate the thorough mixing of the chamber environment. Pollutant spikes were also measured at the same time by different sensors of the same model, placed in different parts of the chamber, giving us high confidence that the chamber is well mixed.
The electrostatic precipitator unit that we used removes over 95% of PM2.5 on a single pass through the device. Given the small volume of the test chamber (1m3) and the high flow rate of the ESP air filtration unit (150m3/h), particles in the chamber are quickly removed to below the detection limit in a matter of minutes, as verified in multiple experiments.

Reviewer 5 Report

Excellent work overall! The authors present a thorough investigation comparing the performance of low-cost PM sensors under a controlled laboratory setting for different humidity settings and PM sources. Their results show the applicability of low-cost PM sensors for monitoring exposure in finer granularity.

Some minor typos:

- Line 517: Please correct reference [?]

- Line 566: The word "to" appears twice in "...capacity to to identify..." 

Author Response

We would like to thank the reviewer for this review. The typos have been corrected.

Round 2

Reviewer 1 Report

Dear

The authors responded to my suggestions.

Congratulations!

Reviewer 4 Report

Dear Authors,

The corrections were considered and the paper is accepted in its present form.